# Elastin-targeted nanoparticles delivering doxycycline mitigate cytokine storm and reduce immune cell infiltration in LPS-mediated lung inflammation

**Shivani Arora**, **Narendra Vyavahare***

Department of Bioengineering, Clemson University, Clemson, South Carolina, United States of America

* narenv@clemson.edu

## Abstract

### Background and purpose

Cytokine storm invoked during acute and chronic lung injury promotes alveolar damage and remodeling. The current study shows that degraded elastin-targeted nanoparticles releasing doxycycline (Doxy NPs) are potent in mitigating cytokines storm, migration of immune cells in the lungs, and inhibiting inflammasome pathways in the LPS mouse model.

### Experimental approach

Cytokine storm and lung injury were induced using LPS and elastase in C57BL/6 mice (rodent model for emphysema). The mice were then treated with I.V. Doxy NPs, blank NPs, or Doxy a day before LPS administration. Cytokine levels, immune cell population, and MMP activity were analyzed in broncheo-alveolar lavage fluid (BALF) 4 hours after LPS administration. Additionally, gene expression of IL-6, IL-1beta, MCP-1, NLRP3, Caspase 1 and MMPs were investigated in alveolar cells on day 3 after LPS administration.

### Key results

Doxycycline NPs but not Doxycycline significantly decreased IL-6, TNF-α, IL-23 and were significantly more effective in decreasing the percentage of immune cells in the BALF. This is the first in-vivo study to demonstrate that Doxycycline can effectively inhibit inflamma-some pathways in the lungs.

### Conclusion and implications

IV administration of elastin antibody conjugated Doxycycline-loaded albumin NPs can effectively modulate the local immune environment in the lungs, which is not achieved by IV Doxycycline even at 100-fold higher dose. This novel method of drug delivery can effectively lead to the repurposing of traditional Doxycycline as a potential adjunct treatment for managing the cytokine storm in the lungs in COPD and viral infections.

**Data Availability Statement:** All relevant data are within the paper and its Supporting Information files.

**Funding:** This work is funded by Elastrin Therapeutics Inc. as a subcontract to Clemson University, Elastrin-002.

**Competing interests:** Elastin targeted nanoparticle system is licensed to Elastrin Therapeutics Inc. and NRV holds significant equity in the company; however, this does not alter our adherence to PLOS ONE policies on sharing data and materials.

## Introduction

Airway remodeling, a most apparent consequence of acute and chronic lung conditions like acute viral infections, including SARS-CoV-2 and emphysema, is a sequel of proinflammatory intra- and extracellular events in the architectural components of the respiratory system [1, 2]. These airway disorders cause epithelial cell apoptosis and inflammatory cellular infiltration. All these pathological events either lead to/or result from elastin damage in the alveoli, which is a critical component of breathing mechanics responsible for lung elastic recoil.

The proinflammatory cytokines; IL-6, IL-1β, TNF-α, IL-23; and proteases; MMPs 2,9 &12; are the common orchestrators of inflammation-associated changes in lungs [3]. They have been shown to decrease elastin mRNA expression or directly degrade elastin in the lungs. These cytokines are currently the prime targets for not only COPD drug discovery but also for acute lung injury occurring from viral infections in the lungs. FDA has approved two classes of IL-6 inhibitors for the treatment COVID 19, viz. a) anti-IL-6 receptor monoclonal antibodies (mAbs) (e.g., sarilumab, tocilizumab), and b) anti-IL-6 mAbs (i.e., siltuximab). Tocilizumab has recently been approved by FDA on Dec.21, 2022 for use in COVID 19 patients as an adjunct therapy along with dexamethasone, while Sarilumab has not yet been approved for cytokine storm management [4]. However, these drugs have severe side effects, including neutropenia, hypofibrinogenemia and increased risk of secondary infections like tuberculosis, bacterial and fungal infections as well as bowl perforation, limiting their use to a generally healthy population [5].

Doxycycline (Doxy) has been found to be a useful drug in many conditions. In a recent report, both pre and post-treatment Vero E6 cells with Doxy were shown to effectively inhibit SARS-CoV-2 strain (IHUMI-3) in a dose-dependent manner [6]. Doxy's anti-inflammatory potential is being harnessed for treatment of chronic conditions including- Brucellosis spondylitis, traumatic brain injury, abdominal aortic aneurysms, where in it has been shown to decrease levels of systemic inflammation markers, IL-6, and inhibit chemokines like MCP-1 and MMPs [7–10].

On the contrary, a recent study by Butler et al have shown that treatment with doxycycline (given orally) does not reduce the recovery time as well as the number of deaths associated with COVID-19 [11]. However, this was systematically given a drug with a substantial loss in first-pass metabolism, so high concentrations in local lung tissue may not have been achieved.

Previously, we have shown that a single i.v. injection of elastin antibody conjugated doxycycline loaded bovine serum albumin nanoparticles (Doxy NPs) can effectively target emphysematous lungs and result in sustained release of Doxycycline locally in lungs over four weeks leading to a decrease in Matrix Metalloproteinases (MMPs) activity as compared to Doxycycline IV [12]. Our approach delivers doxycycline to the lung as nanoparticles go to the site of elastin damage in the lungs, Thus, a small amount of the drug is more effective than a systemic dose. We are not suggesting that such therapy would recover the lungs of COVID patients but highlighting that targeted delivery is better than systemic drug administration.

Doxy acts in part by inhibiting protein kinase B (AKT) signaling pathway and mitogen-activated protein kinases (MAPKs) signaling proteins, including extracellular signal-regulated kinase (ERK), c-Jun amino-terminal kinases (JNK) in vitro in VSMCs [5]. Its inflammasome inhibitory capacity was found to be beneficial in altering the tumor microenvironment in prostate cancer (PC3) and a lung cancer cell line (A549) [13]. However, it is still unknown how Doxy inhibits inflammatory response, particularly in the lungs. We wanted to test if IV delivery of Flexibzumab conjugated doxycycline loaded bovine serum albumin nanoparticles

(Doxy NPs) can prevent acute cytokine storm in the lungs in exogenous administration of endotoxin lipopolysaccharide (LPS) mouse model.

## Material and methods

### Synthesizing doxycycline loaded BSA nanoparticles (Doxy NPs) and BSA nanoparticles (Blank NPs)

Doxy NPs were prepared as per the procedure described by Parasaram et al. [12]. Briefly, 25 mg of Doxycycline Hyclate (Sigma Aldrich, St. Louis, MO) was dissolved along with 100 mg of bovine serum albumin (BSA) in 2 mL of water and was stirred at 500 rpm. Using an automated dispenser, ethanol (4 mL) was added to the Doxy and BSA solution at 1ml/min. Following alcohol addition, 8% glutaraldehyde (40μg/mg BSA) was used to crosslink the particles, and the mixture was stirred for additional 2 hours at room temperature at 500 rpm. The resulting particles were pelleted by centrifuging at 14000 rpm for 10 min. and washed three times with DI water and resuspended in either 1 ml of PBS (for conjugation) or in 1 ml of 5% sucrose (to calculate yield and release).

Blank nanoparticles were prepared similarly, except that no doxycycline was used during the process.

### Conjugation of elastin antibody to nanoparticles

We have created a new humanized elastin antibody that targets degraded elastin named Flexibzumab [14–16]. Albumin NPs were activated with heterobifunctional crosslinker α-maleimide-ω-N-hydroxysuccinimide ester poly (ethylene glycol) (Maleimide-PEG-NHS ester, MW 2000 Da, Nanocs Inc., NY, USA) to achieve a sulfhydryl-reactive particle system. Thus, 2.5 mg of Maleimide-PEG-NHS ester solution was added to 10 mg albumin NP dispersion. The reaction was continued for 1 hour at room temperature. Separately, Flexizumab was thiolated using Trout's reagent (34 μl to 10 μg of antibody in 400μl of HEPES and incubated at room temperature for one hr. After incubation, the Trout's reagent was removed using a 30 KDa filter. The thiolated antibody and pegylated nanoparticles were incubated at 4˚C overnight. The conjugated particles were pelleted at 10,000 rpm for 10 min and resuspended in 5% sucrose by sonication followed by lyophilization for final use.

### Characterization of Doxy NPs

The Doxy NPs made using the above method were analyzed for size, yield, loading percentage, and release profile. The size of the doxy particles was measured using 90Plus Particle Size Analyzer (Brookhaven Instruments Co, Holtsville, NY). The supernatant obtained from the washout during NP preparation was used to estimate the amount of free Doxy by measuring absorbance at 273 nm using a UV spectrophotometer (BioTek Instruments Inc., Winooski, VT). The difference between the initial Doxy used and Doxy in supernatant gave the amount of doxycycline encapsulated in NPs.

### In vitro release

NPs (30 mg) were resuspended with sonication in 1 ml of PBS. After 24 hrs, NPs were centrifuged at 10,000 rpm for 10 min and 3 aliquots of 100μl of the supernatant were used to measure Doxy at 260 nm. Pure doxycycline hyclate solution in PBS was used for making standard curves. Blanking was performed with PBS, and for normalization, we used an equivalent amount of blank BSA NPs that were prepared similarly.

## Lung Injury using elastase and LPS

All procedures involving mice were performed according to the IACUC standards following ethics approval by the animal ethics committee at the Clemson University (Protocol#-AUP2019-040). 6 weeks old Male C57BL6/J mice were purchased from Charles River and were maintained on a chow diet, in pathogen-free cages, and on std. 12h light-dark cycle at 23˚C throughout the study. All experiments on mice were performed at 8 to 10 weeks of age. On Day 1, the lungs of animals in all groups except for the sham group animals were treated with elastase (1.5U, I.T.), and the sham group animals received sterile PBS. On day 8, NP treatment was performed. The animals were divided into four groups: control group (PBS), Doxy IV group (10 mg/kg Doxy), Doxy NPs treated group (10 mg/kg of antibody conjugated Doxy loaded BSA nanoparticles), and blank NPs (10 mg/kg antibody conjugated blank BSA NPs). The drugs were suspended in PBS to formulate IV injections and given via tail vein. One day after NP injections, (day 9) Lipopolysaccharides from Escherichia coli O111:B4 (LPS. Sigma, 1mg/kg I.T.) was administered to all the animals except for the animals in the sham group, which received sterile PBS.

Separate cohorts of 4 experimental groups; control group, Doxy IV group, Doxy NPs group, and blank NPs treated group; were set up for cytokine analysis (n = 8 per group), MMP activity analysis for zymography (n = 5 per group) in BALF and samples were collected 4 hrs after LPS administration on day 9. Separate cohort of experimental groups was set up for gene expression analysis (n = 5 per group) in the alveolar tissue and immune cell population analysis in BALF (n = 6), samples from these cohorts were collected on Day 12 i.e. 3 days after LPS administration. The timeline for the experiment is given in Fig 1. Intratracheal and intravenous drug administration was performed under Isoflurane anesthesia.

## Sample collection for analysis

For sample collection, either after 4 hrs or on day 3, the animals were euthanized using carbon dioxide asphyxiation. The bronchoalveolar lavage fluid (BALF) and the cells were collected as per the protocol described by Hoecke et al. [17]. Briefly, after euthanizing the animal, the tracheas were exposed by making an incision in the neck, and the catheter (made with 23G needle) was sutured inside the trachea for performing the lavage. The lungs were lavaged with 3

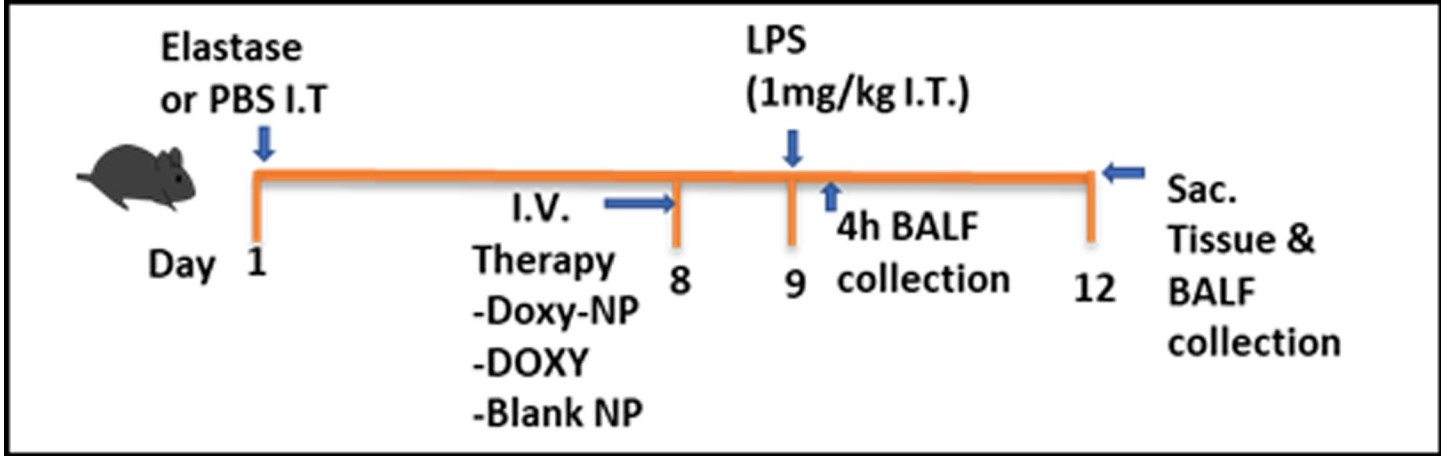

**Fig 1. Experimental timeline.** Separate cohorts of C57 bl/6 mice were used for the study. On day 1, animals in the control group received PBS, whereas all other treatment group received elastase intratracheally. On day 8, therapy was given intravenously (tail vein), and on day 9, LPS was administered intratracheally. The first cohort of animals was sacrificed for sample collection 4 hrs after LPS administration, and the animals in the second cohort were euthanized 3 days after LPS treatment (at Day 12).

ml of PBS containing 100 μm of EDTA, and the lavage fluid was collected and centrifuged for 7 min at 400g and 4°C. Protease inhibitor was added to the supernatant and was stored immediately at -80°C for ELISA and zymography. The cell pellet obtained after centrifugation was treated with ACK lysis buffer at RT for 2 min, and after the ACK was immediately resuspended in PBS. These cells were used for immune population analysis.

## Measurement of cytokine levels

Cytokine levels were measured in BALF samples collected 4 hours after LPS administration to perform cytokine level analysis. The levels of IL6, TNF-α, IL-23 and IFN-γ were measured using kits from Biolegend (Cat. No. 431304, 430904,433704, 430804 resp.) following the manufacturer defined protocol.

## Immune cell population analysis

Immune cell population analyses were performed in the samples collected after three days. The cells from the BALF were collected as described above and, after treatment with ACK lysis buffer, were washed and resuspended in PBS for counting. 1X10^6 cells were used for staining. Fig 4A describes the cell staining and identification strategy.

   Total cells were gated to exclude doublets; from the singlets, live cells were identified using Dapi and were defined as DAPI-ve; from the live cell population, Cd11C low cell population was further narrowed down to identify B cells defined as CD19+MHCII+; T cells; defined as CD3+MHCII-. The remaining cell population, i.e CD19-CD3-MHCII-; was further gated to identify eosinophils defined as, Cd11b+Ly6G-, and neutrophils defined as, Cd11b+Ly6G+.

   In a parallel experiment the same sample and number of cells was used to identify dendritic cells macrophages. Total cells were gated to exclude doublets; from the singlets, live cells were identified using Dapi and were defined as DAPI-ve; from the live cell population, Cd11CHi cell population was further narrowed down to identify Macrophages; defined as SiglicF$^{Hi}$MHCII- and SiglicF$^{int}$MHCII-; and dendritic cells defined as SiglicF-MHCII+. Data was analyzed using FlowJo software (v10.8.) and statistical analysis was performed between groups following one way-ANOVA using GraphPad Prism software The results are represented as Mean±SEM, and a P-value of < 0.05 was considered significant.

   Antibodies used for analysis- anti-mouse CD11c PerCP (cat#117325, biolegend), Anti-mouse MHC Class II (I-A/I-E)-PE-Cy7 (cat#25-5321-82, Invitrogen), Anti mouse Siglec F-PE (Cat#12-1702-80, Invitrogen), DAPI (Cat#422801, Biolegend), TruStainFcX Plus (Cat#156604, biolegend), Anti mouse CD11b-APC (Cat# 101212, biolegend), anti-mouse CD19 Alexa fluor 488 (cat # 115521, biolegend) anti-mouse CD3-Alexa flour 488 (cat# 100210, biolegend), anti-mouse Ly-6G Alexa Flour 700 (cat #127621, biolegend), DAPI (Cat#422801, Biolegend), TruStainFcX Plus (Cat#156604, biolegend).

## MMP activity analysis using zymography

MMPs activity in the BALF harvested 4 hours after LPS treatment was measured using in-gel zymography, following protocol from Leber et al. [18] with slight modifications. Briefly, BALF samples from each treatment group (containing 20 μg of total protein) were loaded on gelatin impregnated SDS gels (ZY00105, Fisher Scientific). Coomassie blue staining was used for visualizing the sites of proteolytic digestion.

## RNA extraction and quantitative reverse transcriptase PCR

The expression of genes of interest was performed in lung samples harvested on day three after LPS treatment. The lung tissue was stored immediately after retrieval in the Trizol. The tissue was homogenized in Trizol using Powergen 125 (FS-PG125, Fisher Scientific) homogenizer, and RNA was extracted using the Zymogen total RNA microprep kit (Zymogen) according to the product instructions. A Nanodrop instrument was used for quantitative and qualitative analysis of the extracted RNA. cDNA synthesis was performed using 200ng of total RNA, iScript gDNA clear cDNA synthesis Kit (Cat# 1725034, Bio-Rad) was used for this. The total cDNA obtained was diluted five times before amplification with iTaq- Universal SYBR Green Supermix reagent (Cat# 172512, Bio-Rad) on Bio-Rad quantitative PCR platform (96-well format). Quantification was performed using the delta-delta $C_T$ method, using HPRT as a housekeeping gene. Details of the primer sequence are given in Table 1.

## Quantification and statistical analysis

The number of biological replicates (n) for each experiment is indicated in the figure legend. Statistical analysis was performed between treatment groups using one-way ANOVA as there were more than three groups in the study, with corrections for multiple testing. The results are represented as Mean±SEM, and a P-value of $\leq 0.05$ was considered significant.

## Results

The current study aimed to compare the efficacy of targeted Doxy NPs with standard doxycycline IV treatment in mitigating cytokine spurt as well as at evaluating the mechanism of action of Doxycycline by which it exerts its anti-inflammatory action in the lungs.

### Doxycycline NPs but not doxycycline IV mitigates cytokine spurt in BALF

We analyzed the levels of proinflammatory cytokines-IL-6, TNF-α, IL-23 and IFN-γ in the BALF, 4 hours post LPS treatment using ELISA (Fig 2). Our results indicate that at the doses used in the study, Doxy NPs were significantly more effective in lowering IL-6, TNF-α, and IL-23 levels (with a p value<0.0001), whereas treatment with Doxy IV was ineffective in lowering the level IL-23 & TNF- α and only slightly decreased IL-6 levels (p = 0.012) (Fig 2A–2C). Levels of IFN-γ were significantly reduced by both Doxy NPs and IV Doxy (p = 0.0074 and p<0.0001). However, with the Doxy NPs, we observed that they were maintained in a range closer to the control group (71.38±9.72pg/ml vs. Control 59.80±4.25pg/ml). In contrast, its level was significantly lowered compared to controls by Doxy IV treatment (19.95±5.04pg/ml, with p<0.0001) (Fig 2D). These results indicate higher efficacy of Doxy NPs over Doxy IV treatment.

### Doxycycline nanoparticles are equally effective in decreasing MMPs expression and activity

Previously we showed that MMPs are inhibited with Doxy NPs in the elastase model of emphysema; we thus evaluated whether pretreatment with Doxy NPs could inhibit MMP activity compared to Doxy treatment in the LPS model.

We analyzed protein level activity in BALF (by gelatine zymography) and gene expression of MMP 2, 9 &12 in the lungs (by RT-PCR). Both Doxy IV and Doxy NPs treatments showed an 8-9-fold decrease in MMP-9 activity (Fig 3A and 3B). We found that Doxy NPs treatment was slightly more efficient than doxy IV in inhibiting MMP-9 activity (with p = 0.0002, Fig 3B) as well as expression of MMP-2, MMP-9 and MMP-12 in the lungs (Fig 3C).

**Table 1. Details of primers used in study.**

| Target Gene | Primer Sequence |
|---|---|
| IL6 | F- ACCACGGCCTTCCCTACTTC |
| | R- TTGGGAGTGGTATCCTCTGTGA |
| IL1β | F- CACAGCACATCAACAAG |
| | R- GTGCTCTAGTCCTCATCCTG |
| MCP1 | F- GTCTGTGCTGACCCCAAGAAG |
| | R- TGGTTCCGATCCAGGTTTTTA |
| MMP 2 | F-CACACCAACACTGGGACCTG |
| | R- AGAATGTGGCCACCAGCAAG |
| MMP 9 | F- GCTGACTACGATAAGGACGGCA |
| | R- TAGTGGTGCAGGCAGAGTAGGA |
| MMP 12 | F- TTTCTTCCATATGGCCAAGC |
| | R- GGTCAAAGACAGCTGCATCA |
| HPRT | F- CTGGTGAAAAGGACCTCTCG |
| | R- TGAAGTACTCATTATCAAGGGCA |
| NLRP3 | F- CTGCGGACTGTCCCATCAAT |
| | R- AGGTTGCAGAGCAGGTGCTT |
| Caspase-1 | F-5′–AACCAGGAGAATGTTTCCAACCT–3′ |
| | R-5′–AAACACCAGGCCAAGCTTCTT–3 |
| 1L-18 | F- 5′–ATGGCTGCTGAACCAGTAGAAG–3' |
| | R- 5′–CAGCCATACCTCTAGGCTGGC–3 |
| β-Actin | F- 5′–CGATGCCCTGAGGGTCTTT–3' |
| | R:5′–TGGATGCCACAGGATTCCAT–3 |

## Doxy NPs decrease immune cell infiltration In BALF

Having seen a decrease in the cytokine levels and MMP activity in the BALF, we were further interested in evaluating whether doxycycline treatment can inhibit immune cell infiltration in the lungs. We analyzed the percentage of B cells, T cells, eosinophils, neutrophils, dendritic cells, and macrophages in the BALF using FACS (Fig 4).

We observed that both Doxy IV and Doxy NPs significantly brought down the percentage of B cells (from 17.04% in the blank NP treated group to 5.287% & 2.106% in the Doxy IV and Doxy NPs treated groups resp.) (Fig 4B), T cells (from 33.84% in the blank NP treated group to 16.64% & 8.54% in the Doxy IV and Doxy NPs treated groups resp.) (Fig 4C), neutrophils (from 23.13% in the blank NP treated group to 16.29% & 9.2% in the Doxy IV and Doxy NPs treated groups resp.) (Fig 4D) and Dendritic cells (from 4.15% in the blank NP treated group to 1.44% & 0.62% resp.) (Fig 4I).

However, the increased percentage of eosinophils was not affected by Doxy IV treatment (from 0.4% in controls to 11.38% and 7.88% resp.). This increased eosinophil count was mitigated only by the Doxy NPs treatment (2.84%) (Fig 4E).

We also observed that the overall percentage of macrophages was also brought down by Doxy NPs. Of particular interest here is the macrophage population defined as SiglicF$^{Hi}$ (macrophages that have high Siglic F receptor density) that was particularly affected by doxycycline treatment [19]. A significant decrease was seen in their population with Doxy IV and Doxy NPs treatment resp. (p<0.05 as compared to controls) (Fig 4F–4H).

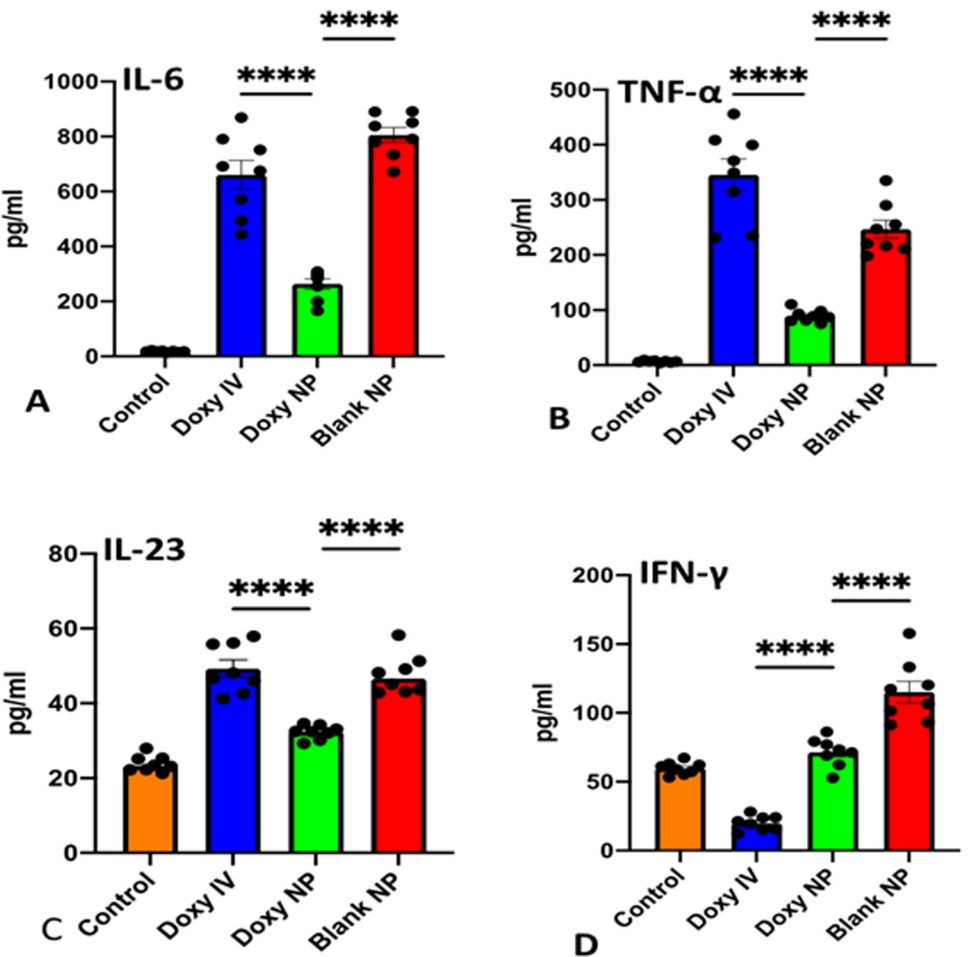

**Fig 2. Doxycycline nanoparticles more effectively mitigate cytokine storm.** Elastase and LPS were administered I.T, and the drug treatment was given by the IV route. 4 hours after LPS administration BALF and lungs were harvested for analysis. (A) to (D) **ELISA** of IL-6, TNFα, IL-23 and IFN-γ resp. in BALF, showing Mean± SEM of n = 8 mice per group, *p < 0.05, one-way ANOVA.

## Doxycycline Inhibits NLRP3 and effectors of inflammasome pathway

To evaluate the mechanism of Doxycycline in lung inflammation, we focused on its ability to influence the expression of the effectors of inflammasome pathways. We performed qPCR for NLRP3, Caspase 1, IL-6, IL-1β, and MCP-1 in the lung alveolar cells. We observed a significant decrease in IL-1β (p = 0.0029), and MCP-1 (p = 0.015) expression in the Doxy NP treatment group whereas Doxy IV treatment failed to cause any significant changes in their expression. However, both were equally effective in significantly lowering the expression of IL-6, NLRP3 and Caspase 1 (Fig 5A–5E). We also performed IHC for NLRP3 expression in lungs (S2 Fig) and observed a significant decrease in its expression with Doxy NP treatment, however since we did not quantify the IHC expression and thus we refrain from commenting on the actual status.

## Effective doxycycline dose in Doxy NPs vs. Doxy IV groups

We have previously established that Doxy loading in BSA NPs is about 17%, and Doxy NPs release doxycycline slowly over 30 days [12]. We confirmed in the current study (S1 Fig) that the amount of doxycycline released in 24 hrs. from the dose of 10 mg/kg of Doxy NPs is

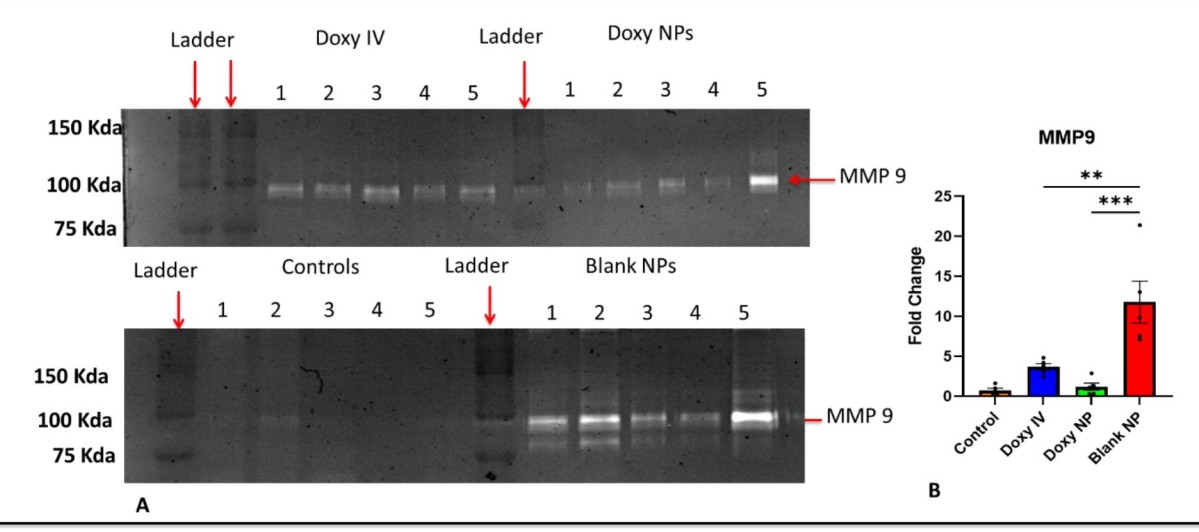

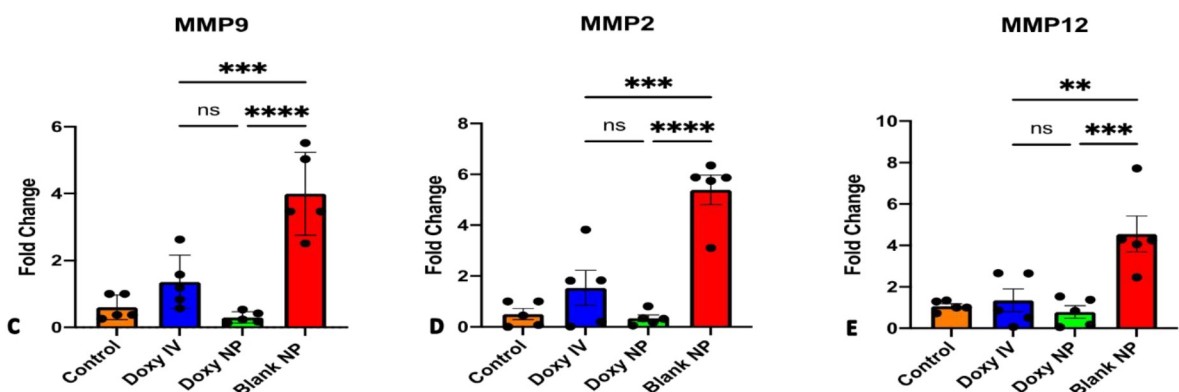

**Fig 3. Doxycycline nanoparticles effectively inhibit MMP 2 & 9.** A) In-Gel Zymography was performed to detect MMP activity in the BALF samples. (B) Quantification of the enzymatic activity was done using Image J software, results represented as Mean± SEM; n = 5 mice per group; *p < 0.05, as analyzed by one-way ANOVA. (C)-D) qPCR based relative quantification of MMPs expression in the lungs harvested at day 12 from each treatment group; HPRT was used as housekeeping gene; results showing fold change with respect to HPRT (the house keeping gene); are represented as Mean± SEM of n = 5 mice per group, *p < 0.05, **p < 0.01, as analyzed by one-way ANOVA.

~90 μg/kg. This is about 100 times lower dose than what is available for std. Doxy IV (10 mg/kg, with its half-life of 16-22hrs) used in the current study. Thus, Doxy NPs were proven to be significantly more potent in inhibiting proinflammatory cytokines, macrophage infiltration, and eosinophilia in lungs than std. Doxy IV administration at 100 fold lower concentrations.

## Discussion

As much as there is encouraging data to support Doxycycline testing for inflammatory lung conditions, there are also some conflicting reports that indicate against doxycycline use. It has been reported that doxycycline hyclate aggravates granulomatous lung inflammation in a dose-dependent manner in S. mansoni-infection by causing eosinophilic infiltration and by downregulating Th2 effectors [20]. In another clinical trial, doxycycline was found to be ineffective in lowering MMP-8 and 9, IL-6, and IL-8 concentrations and in improving lung function parameters; however, it did decrease systemic inflammation in COPD patients [21]. The possible justification given for these contradictions is the variation in the percentage of

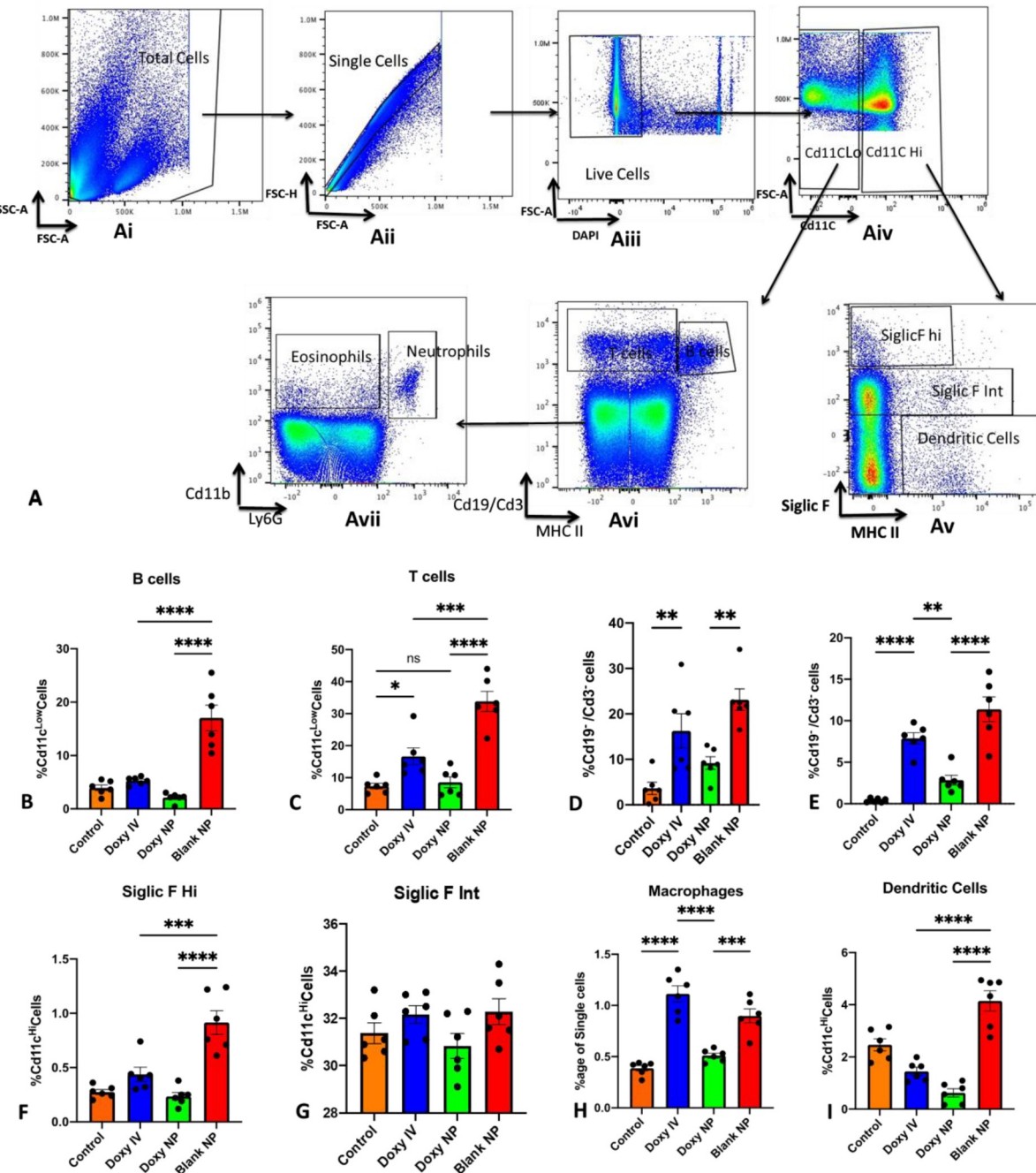

**Fig 4. Doxycycline nanoparticles more effective in mitigating immune cell recruitment to lungs.** (A) Gating Strategy for FACS based analysis of immune cell population in BALF, (B-I) Quantification of B cells(B), T cells(C), Neutrophils(D), eosinophils (E), SiglicF$^{hi}$ macrophage population (F), SiglicF$^{int.}$ macrophage population (G), total macrophages (H), and dendritic cells (I) in all treatment groups. Results showing Mean± SEM of n = 6 mice per group, *p < 0.05, **p < 0.01, ***p < 0.001 one-way ANOVA.

neutrophil infiltration seen at the beginning of the study and the possibility that doxycycline has an altogether different mode of action in the lung as compared to the aorta or periodontal diseases.

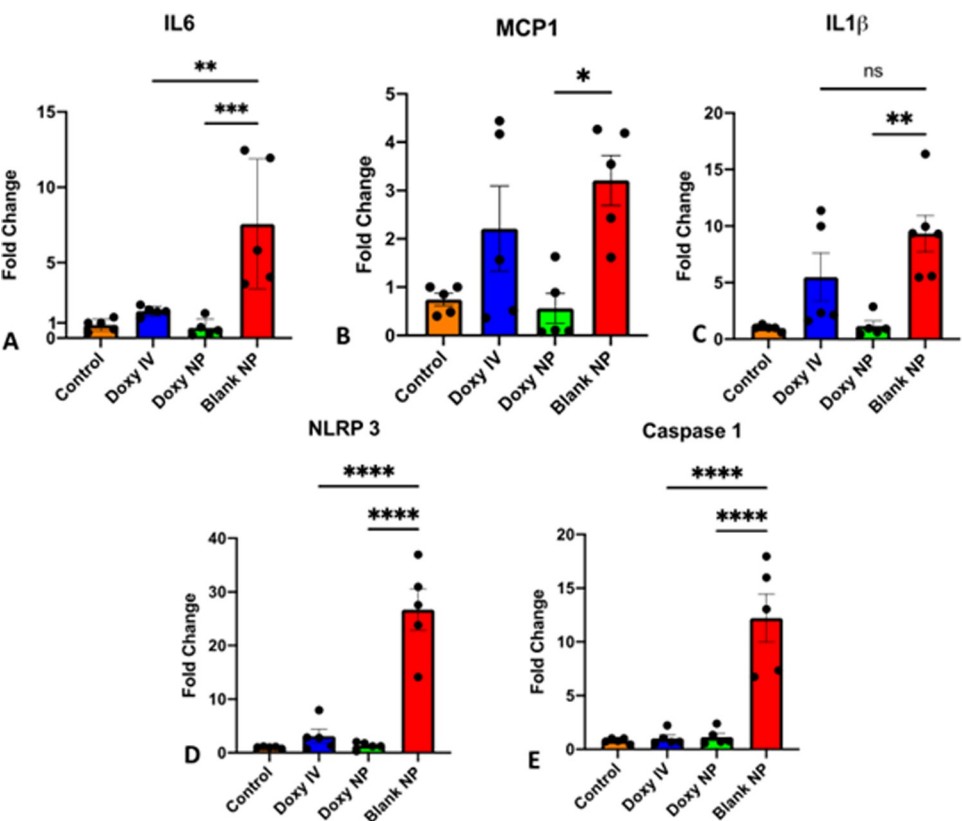

**Fig 5. Doxycycline nanoparticles inhibit Inflammasome formation.** A)-D) qPCR based relative quantification of IL-6, MCP-1, IL1β, NLRP3 and Caspase 1 expression resp. in the lungs harvested at day 12 from each treatment group; β Actin was used as housekeeping gene; results showing Mean± SEM of n = 5 mice per group, **p < 0.01, ****p < 0.0001as analyzed by one-way ANOVA.

Morphological lung changes precipitated by either viral or chronic autoimmune disorders incite inflammatory responses. IL-6, one of the most prominent proinflammatory cytokines unleashed by either acute or chronic lung injury, was examined in the study. It participates at multiple levels to induce cellular damage. A) It activates multiple downstream signaling pathways, including Janus/kinase signal transducers and activated transcription JAK-STATA P13K-AKT(STAT 1,3 and 5) Rapidly accelerated fibrosarcoma RAS-RAF, SRC-yes associated protein (YAP)-Neurogenic locus notch homolog (NOTCH), via classical signal transduction. B) IL-6 not only causes recruitment of immune cells in the local tissue environment but also helps in the direct proliferation of B cells, activates cytotoxic T cells, prompts helper T cells, and causes induction of proinflammatory Th-17 cell responses. In the present study with Doxy NPs, we were able to mitigate IL-6 levels in BALF and turn down its expression in the lungs.

IL-23 is another proinflammatory cytokine of interest that was investigated in the current study. It is involved in the differentiation of Th17 cells in a proinflammatory context and especially in the presence of TGF-β and IL-6. Activated Th17 cells produce IL-17A, IL-17F, IL-6, IL-22, TNF-α, and GM-CSF. Inflammatory macrophages express IL-23R and are activated by IL-23 to produce IL-1, TNF-α, and IL-23 itself. These effects identify IL-23 as a central cytokine in autoimmunity and a highly promising treatment target for inflammatory diseases [22]. We observed a significant decrease in IL-23 levels in the lavage fluid after Doxy NP treatment. IFN-γ is responsible for macrophage activation in alveolar tissue and inducing autophagy. However, increased levels of IFN-γ are considered beneficial when it comes to countering the

viral attack, and specifically counter acts LPS-induced immune activation by suppressing TLR4-mediated gene activation [23–25]. Hence, a therapy that could fine-tune its level would be more appropriate in lung pathology. In our study, we observed that Doxy NPs did bring down the levels of IFN-γ to normal levels but did not mute it entirely as Doxy IV group. TNFα is another cytokine well established for its role in promoting lung inflammatory conditions. It is responsible for the activation of NF- κB and Activator Protein 1 (AP1) pathway and the production of ICAM-1, VCAM-1, and RAGE [26]. In this study with Doxy NPs treatment, we observed a significant decrease in the TNF-α levels.

In the current study, we modified the pre-established method of inducing lung injury to induce a cytokine storm. Lungs of C57BL/6 mice were first treated with elastase (I.T.), and seven days after cytokine infiltration was induced by IT LPS. It mimics compromised lung conditions for COPD patients. The priming of lungs with elastase not only induces damage to the elastin rich connective network of the lungs but also causes infiltration of neutrophils [27]. The second hit with LPS was used to aggravate the condition further so that significantly higher levels of proinflammatory cytokines and immune cell infiltration could be achieved, as is seen clinically during asthmatic attack or in acute viral infections. This model has been previously used by many researchers, including us [12, 28–31], to mimic the pathology of emphysema. Using this approach, we were not only able to achieve very high concentrations of proinflammatory cytokines IL-6 (up to 805.74±72.06pg/ml), TNF-α (246.77±43.31pg/ml), IL-23 (47.70±4.91pg/ml) and IFN-γ (115.05±20.88pg/ml) but also significantly higher immune cell infiltration (B cells, T cells, neutrophils, macrophages and eosinophils) as compared to control group (all at p<0.0001).

Having established the model, we further analyzed the mechanism of action, as well as the efficacy of elastin-antibody, conjugated Doxycycline NPs, to evaluate their efficacy for alleviating cytokine storm over Doxycycline IV to mimic acute and chronic inflammatory lung disorders, including COPD and SARS CoV 2 infections. We compared the efficacy of Doxy NPs with standard Doxycycline IV. Of note, we have already shown using dye loaded nanoparticles, that elastin antibody targeted nanoparticles specifically target damaged alveolar lung elastin when given i.v. and are retained in lungs over a period of 2–4 weeks [12]. We found that reformulating Doxycycline to nanoparticles not only helps in targeting it to the desired site of action but also makes the drug more effective in mitigating immune responses at 100 times lower concentrations.

In the current study, we observed that although Doxycycline in either of the formulations (std IV or NPs) was capable of decreasing dendritic cell, B Cell, and T cell population, However only when targeted in the form of Doxy NPs, was it capable of mitigating macrophage, neutrophil and eosinophil count as well as alleviate the cytokine storm. This selective decrease in Siglic F$^{hi}$ macrophage, neutrophil and eosinophil population caused by Doxy NPs could be accounted responsible for its substantial response against cytokine spurt. Siglic F$^{hi}$ macrophage population has been defined as therapeutic target for treating acute liver injury [32]. Many novel therapeutic strategies are also focusing on alleviating eosinophilia during COPD. Mepolizumab monoclonal antibodies directed against IL 5 to mitigate eosinophil population in lungs is currently under phase 3 clinical trial for the treatment of COPD. Although using MAbs is an attractive strategy, it has several limitations that restrict its use in a limited population [11, 33, 34]. Targeted delivery of a pre-approved drug (Doxycycline) to lungs to check eosinophilia and cytokine spurt offers an attractive opportunity to fast-track the translation to the clinic.

We also found that high doses of Doxy IV can (10 mg/kg IV) inhibited MMPs activity and recruitment of immune cells but was significantly less effective in lowering cytokine levels as well as immune cell infiltration, in the LPS primed lungs as compared to Doxy NPs, that were

specifically targeted to damaged elastin in lungs. In a recent report, IV treatment with doxycycline hyclate was reported to cause dose-dependent eosinophilia by downregulating Th-2 responses during S. mansoni-infection [20]. In vitro studies with low doses of doxycycline (20 to 40 mg) were shown to inhibit IL-6, but an inhibitory effect on MMPs was only observed when it was used in higher doses of 100–120 mg [35]. The results of our study are in line with previous findings [36] and provide a unique method for achieving a slow, sustained release of a low dose of doxycycline that not only inhibits inflammatory response but also lowers MMP expression.

Although it has been previously established that doxycycline inhibits proinflammatory responses by acting via the mTOR pathway in cancer cell lines, its impact on the inflammasome pathway, particularly in the lungs, has not yet been evaluated. There have been only a few in-vitro studies conducted that report the inflammasome inhibiting potential of doxycycline. Xu et al. have tested its inflammasome inhibiting potential in-vitro against P. gingivalis, the causative organism for gingivitis [37], and Alsaadi et al., have reported that in cancer cell lines, doxycycline is capable of suppressing NLRP3 derived inflammatory signals due to which it attenuates cancer cells growth [13]. From these studies, it is indicated that the anti-inflammatory activity of doxycycline could be partly attributed to its inflammasome pathway inhibition. However, a definite in-vivo study to evaluate its activity in this particular context is lacking to date.

The use of NLRP3 inhibitors for treating chronic inflammatory conditions in the lungs has been theorized by many groups [38, 39]. NLRP3 mediated inflammasome formation activates caspase-1 mediated programmed cell death and is responsible for not only increased neutrophil infiltration but the upregulation of chemokines and proinflammatory cytokines via this pathway also leads to eosinophilia [40] as well as increased housing of macrophages, B cells and T cells in lungs. Inhibition of NLRP3 by doxycycline in our study explains how it inhibits infiltration of immune cells and mitigates the cytokine storm.

The current study is the first of its kind to evaluate the action of doxycycline on the inflammasome pathway in vivo in the lungs. We evaluated the effect of doxycycline on the inflammasome pathway three days after induction of injury and four days post-treatment with either Doxycycline IV or Doxy NPs. We observed a significant upregulation of IL-6, MCP1, IL-1β in the injured group. MCP1, IL-1β levels significantly decreased with doxycycline NP treatment, whereas IL-6 levels were effectively brought down by both std. doxy and Doxy Nps. This inhibition of IL-6, MCP1, IL-1β at the alveolar tissue level also explains the improvement in immune cell infiltration seen with Doxy NP treatment.

## Limitations

The study has a few limitations. First, in the current study, we used a low dose of elastase (1.5U/kg) to induce lung injury; at this dose, neither is it possible nor did we perform a lung compliance study to evaluate the physiological outcome. Second, a very high dose of std. doxycycline hyclate (10 mg/kg) was used in the study, translating to an IV dose of 600 mg in an average healthy adult. On the other hand, Doxy NPs only contained a 17% loading of doxycycline and only released ∼90 μg/kg in mice, which means the dose use was 100 times lower than Doxy IV. The use of such a high-dose Doxy IV might have undermined the actual efficacy of targeted Doxy NPs. Third, although the intervention was performed seven days after elastase treatment, it was done 24 hrs prior to LPS treatment. This implies that treatment with Doxycycline NPs were effective against sudden exacerbation of inflammatory response in the lungs with preexisting mild inflammatory conditions. However, for better clinical extrapolation of the results from the current study, using a more robust COPD model with a lower dose of std

doxycycline IV is recommended. We also did not measure the actual concentration of Doxy in BALF or serum but blank NPs that were targeted with elastin antibody did not show any effects, clearly showing Doxy delivered and caused the changes. Third, as Doxy NPs release the drug over 30 days, the long-term effect of this therapy needs to be evaluated. Fourth, the study is the first of its kind to evaluate the inhibitory effect of doxycycline on the Inflammasome pathway in lungs, further studies using a genetic model for NLRP3 K/O mice, and NLRP3 agonists are warranted.

## Conclusion

We show that degraded elastin-targeted nanoparticles can deliver doxycycline to the lungs and prevent cytokine storm and inflammatory cell infiltration at 100-fold lower concentration that Doxy IV treatment. This approach can lead to fast-track target validation via drug repurposing. It can considerably decrease the time lag between bench to bedside therapy to deal with emergencies that develop rapidly across the globe, as the world has recently seen with the novel COVID 19 outbreak. In the future, we will test if one dose of targeted Doxy NP therapy would keep the cytokine storm in check for more extended periods.

## Supporting information

**S1 Fig. Characterization of Doxy NP- doxycycline release profile.**
(JPG)

**S2 Fig. NLRP3 staining in lungs-IHC for NLRP3 expression in lungs.**
(JPG)

**S1 Raw images.**
(ZIP)

## Acknowledgments

We Acknowledge Dr. Agnes-Nagey Mehesz and Dr. Guzelia Korneva from the COBRE Center for their support.

## Author Contributions

**Conceptualization:** Narendra Vyavahare.

**Data curation:** Narendra Vyavahare.

**Formal analysis:** Shivani Arora.

**Funding acquisition:** Narendra Vyavahare.

**Investigation:** Shivani Arora.

**Methodology:** Shivani Arora.

**Project administration:** Narendra Vyavahare.

**Supervision:** Narendra Vyavahare.

**Writing – original draft:** Shivani Arora.

**Writing – review & editing:** Shivani Arora, Narendra Vyavahare.

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
