## [Decision Letter · Decision Letter 0]

27 Dec 2022

PONE-D-22-32324Targeted delivery of doxycycline with nanoparticles mitigates cytokine storm and reduces immune cell infiltration in LPS mediated lung inflammation.PLOS ONE

Dear Dr. Vyavahare,

Thank you for submitting your manuscript to PLOS ONE. After careful consideration, we feel that it has merit but does not fully meet PLOS ONE’s publication criteria as it currently stands. Therefore, we invite you to submit a revised version of the manuscript that addresses the points raised during the review process.

Your manuscript has been reviewed by two experts and we received positive feedback. However, there are multiple points raised by the reviewers , which require your attention during revision.

We look forward to receiving your revised manuscript.

Kind regards,

Partha Mukhopadhyay, Ph.D.

Section Editor

PLOS ONE

Journal Requirements:

2. To comply with PLOS ONE submissions requirements, in your Methods section, please provide additional information regarding the experiments involving animals and ensure you have included details on (1) methods of anesthesia and/or analgesia, and (2) efforts to alleviate suffering.

"Elastin targeted nanoparticle system is licensed to Elastrin Therapeutics Inc. and NRV holds significant equity in the company"

"The work presented here is partially funded by the grant from South Carolina Research Authority to Elastrin Therapeutics Inc. and subcontracted to Clemson University."

“The work presented here is partially funded by the grant from South Carolina Research Authority to Elastrin Therapeutics Inc. and subcontracted to Clemson University.”

Reviewers' comments:

Reviewer's Responses to Questions

**Comments to the Author**

1. Is the manuscript technically sound, and do the data support the conclusions?

Reviewer #1: Yes

Reviewer #2: Yes

2. Has the statistical analysis been performed appropriately and rigorously? 

Reviewer #1: Yes

Reviewer #2: Yes

3. Have the authors made all data underlying the findings in their manuscript fully available?

Reviewer #1: Yes

Reviewer #2: No

4. Is the manuscript presented in an intelligible fashion and written in standard English?

Reviewer #1: Yes

Reviewer #2: Yes

5. Review Comments to the Author

Reviewer #1: Major Comments:

1. There is already a study he Doxycycline for community treatment of suspected COVID-19 in patients. Ref: https://www.thelancet.com/journals/lanres/article/PIIS2213-2600(21)00310-6/fulltext, DOI: https://doi.org/10.1016/S2213-2600(21)00310-6. According to this article – “treatment with doxycycline was not associated with clinically meaningful reductions in time to recovery or hospital admissions or deaths related to COVID-19, and should not be used as a routine treatment for COVID-19.” How do the authors justify this observation with their current manuscript findings? It seems like these preclinical results have no application in clinical settings.

2. Doxycycline is a broad-spectrum antibiotic. It has severe adverse effects on the patients. The authors should justify how the beneficial effect of inhibiting the cytokine storm can outrun the adverse effects of the drug. They should write briefly about this in the Discussion.

3. Why the nanoparticle formulation of Doxycycline is better than Doxycycline? The authors should also justify that.

4. Why were the Doxycycline NPs delivered by IV route? Would it not be effective if it is administered by IP or oral route?

5. Have the authors tested the effect of Doxycycline NPs on the other organs, such as the liver, kidneys, and/or blood cells? Do they have any data on the safety profile of Doxycycline NPs at the reported dose?

Please find my comments below:

1. There are a few grammatical mistakes throughout the manuscript. The authors should address this during the revision.

2. Abstract: ‘BALF’ was not defined.

3. Abstract: In the Conclusion, ‘NPs’ is written as ‘Nps’.

4. Method: In the flow cytometry method, the authors should write the fluorophores used for each marker.

5. Figure 4: Flow cytometry dot plots are not acceptable. The population is behind the axis. The authors should reanalyze this data and report the results accordingly.

Reviewer #2: Arora et al., investigated the mitigating role of Doxy NPs in the LPS mediated lung inflammation. For the benefit of the readers, here are my minor comments for further improvement of this manuscript.

1) In the second paragraph of the introduction section, the authors said that drugs have severe side effects. I request the authors to mention the side effects and cite the relevant work.

2) Supplementary Figure S2 is not mentioned in the manuscript’s main text.

3) I request the authors to provide details of all the primers used in this study.

4) Can the authors comment on the findings of figure 2D where they found that amount of IFN is lower than the control in Doxy IV case? What does this signify?

5) In fig. 3C the authors say fold change on the y-axis, but it needs to be clarified it is with respect to which parameter. I request the authors to clarify it.

6) I request the authors to rearrange the paragraph titled “Doxy NPs decrease immune cell infiltration in BALF”. The current paragraph is not very clear.

7) For the benefit of the readers, I request the authors to mention what SiglicFHi stands for and maybe cite relevant work.

8) In figure 4, the authors have done statistical tests for different groups in different panels. I request the authors to keep everything consistent.

9) In supplementary fig. S1, I request the authors to add the number of sample information. Also, what does the error bar represents?

Possible typos:

1) In page 6, I request the authors to use standard curve instead of std which can be confused with standard deviation.

2) In page 7, the authors mention fig. 3A, but I think it is a typo.

3) In page 11, I think the authors meant Supp Fig. “S1”.

4) The captions in figure 2 should be corrected. I think it should be A to D instead of B to D.

6. PLOS authors have the option to publish the peer review history of their article (what does this mean?). If published, this will include your full peer review and any attached files.

Reviewer #1: No

Reviewer #2: No

---

## [Author Response · Author response to Decision Letter 0]

15 Feb 2023

We appreciate a thorough review of our article by both reviewers. We have made all the requested corrections and discussed/explained wherever needed in the manuscript. We believe the article is stronger after this review.

Reviewer #1: Major Comments:

1. There is already a study he Doxycycline for community treatment of suspected COVID-19 in patients. Ref: https://www.thelancet.com/journals/lanres/article/PIIS2213-2600(21)00310-6/fulltext, DOI: https://doi.org/10.1016/S2213-2600(21)00310-6. According to this article – “treatment with doxycycline was not associated with clinically meaningful reductions in time to recovery or hospital admissions or deaths related to COVID-19, and should not be used as a routine treatment for COVID-19.” How do the authors justify this observation with their current manuscript findings? It seems like these preclinical results have no application in clinical settings.

Response: Thank you for your comment. We agree that systemic DOXY treatment did nor reduce the time of recovery. That is the drawback of systemic administration as you need a large and continuous dose to get the minimum drug at the lung site. Our approach is targeting this drug to the lung by encapsulating it in a nanoparticle and deliver it to the site of elastin damage in lungs, Thus, small amount of the drug is more effective than a systemic dose. This is the advantage of nanoparticle-based drug delivery that allows the local release of Doxy. We are not claiming that we can recover the lungs of COVID patients but highlighting that targeted delivery is better than systemic drug administration. 

2. Doxycycline is a broad-spectrum antibiotic. It has severe adverse effects on the patients. The authors should justify how the beneficial effect of inhibiting the cytokine storm can outrun the adverse effects of the drug. They should write briefly about this in the Discussion.

Response: Again, this is the drawback of systemic delivery of Doxy. We developed targeted therapy, so we need several fold lower dosages than systemic administration with this approach and drug is encapsulated in nanoparticles and delivered at the site. This is now explained in the discussion. 

3. Why the nanoparticle formulation of Doxycycline is better than Doxycycline? The authors should also justify that.

Response: Nanoparticles enclose Doxy and take it to the site of inflammation in the lungs. We have an antibody attached to the nanoparticle that takes these particles to degraded elastin in the lungs. The advantage is that the drug is now delivered locally rather than systemically, which can lead to systemic side effects and need large quantities of the drug to be effective. We now explain this in the discussion. 

4. Why were the Doxycycline NPs delivered by IV route? Would it not be effective if it is administered by IP or oral route?

Response: Nanoparticles cannot be given by oral route as first-pass metabolism and degradation in the stomach and targeting is not possible. IP route also has the same issue of nanoparticles being not effective. IV route is routinely used for drug delivery vehicles and that takes nanoparticles quickly to lungs. 

5. Have the authors tested the effect of Doxycycline NPs on the other organs, such as the liver, kidneys, and/or blood cells? Do they have any data on the safety profile of Doxycycline NPs at the reported dose?

Response: We have already published a paper showing biodistribution of Doxy NPs (Pulm Pharmacol Ther. 2016 Aug; 39: 64–73. We have not seen any toxicity in liver or kidneys with these nanoparticles with histology.

Minor comments:

1. There are a few grammatical mistakes throughout the manuscript. The authors should address this during the revision.

Response: The manuscript has been reviewed, and grammatical errors have been corrected 

2. Abstract: ‘BALF’ was not defined.

Response: BALF has been defined as Broncho-alveolar lavage fluid

3. Abstract: In the Conclusion, ‘NPs’ is written as ‘Nps’.

Response: Correction has been made per the suggestion

4. Method: In the flow cytometry method, the authors should write the fluorophores used for each marker.

Response: A list of antibodies with their respective fluorophores and source has been added in the method section as a separate paragraph at the end under the subheading Immune cell population analysis and reads as given below-

 “Antibodies used for analysis- anti-mouse CD11c PerCP (cat#117325, biolegend), Anti-mouse MHC Class II (I-A/I-E)-PE-Cy7 (cat#25-5321-82, Invitrogen), Anti mouse Siglec F-PE (Cat#12-1702-80, Invitrogen), DAPI (Cat#422801, Biolegend), TruStainFcX Plus (Cat#156604, biolegend), Anti mouse CD11b-APC (Cat# 101212, biolegend), anti-mouse CD19 Alexa fluor 488 (cat # 115521, biolegend) anti-mouse CD3-Alexa flour 488 (cat# 100210, biolegend), anti-mouse Ly-6G Alexa Flour 700 (cat #127621, biolegend), DAPI (Cat#422801, Biolegend), TruStainFcX Plus (Cat#156604, biolegend)”

5. Figure 4: Flow cytometry dot plots are not acceptable. The population is behind the axis. The authors should reanalyze this data and report the results accordingly.

Response: We think the reviewer is questioning Fig 4Aiii and Fig 4Aiv. In response to this comment, please find below our justification:

We have used The Sequential/Successive Gating strategy to identify the population of interest. For this purpose, with respect to “cell count,” we must choose the starting point on X and/or Y axis as “0” meaning 0%. We have followed this rule where we have measured cell count in Fig 4Ai, Fig 4Aii (on both X and Y axis); on the Y axis in Fig Aiii, and on the X axis in Fig 4Aiv. By doing so, we have tried to make the most meaningful representation of the data that we have collected.

Further, we used FlowJo Version 10.8.1, and we have placed gates on the axis, by using this strategy we have coved the axis that falls behind the gates as the software sees that population and takes it into account while doing the analysis and therefore the analysis did not change when we redid the analysis.

Therefore, we refrain from making the changes in the revised manuscript and request the reviewer to consider the explanation given above for our choice of not changing the original analysis.

Reviewer #2:

 Arora et al., investigated the mitigating role of Doxy NPs in the LPS mediated lung inflammation. For the benefit of the readers, here are my minor comments for further improvement of this manuscript.

1) In the second paragraph of the introduction section, the authors said that drugs have severe side effects. I request the authors to mention the side effects and cite the relevant work.

Response: Per the review suggestion we have added details on the adverse effects related to the drugs as well as updated the current status of the drugs. We have also cited relevant literature and resources for the same. The following text has been added in the revised version-

Tocilizumab has recently been approved by FDA on Dec.21, 2022 for use in COVID 19 patients as an adjunct therapy along with dexamethasone, while Sarilumab has not yet been approved for cytokine storm management (COVID 19 Treatment guidelines by NIH) However, these drugs have severe side effects, including neutropenia, hypofibrinogenemia and increased risk of secondary infections like tuberculosis, bacterial and fungal infections as well as bowl perforation, limiting their use to a generally healthy population (Charan et al 2021).

2) Supplementary Figure S2 is not mentioned in the manuscript’s main text.

Response: We have included the description of Fig S2 in the main text on Page 11, and it reads as-

“We also performed IHC for NLRP3 expression in lungs (Fig S2) and observed a significant decrease in its expression with Doxy NP treatment, however since we did not quantify the IHC expression and thus we refrain from commenting on the actual status.” under the subheading “Doxycycline Inhibits NLRP3 and Effectors of Inflammasome Pathway” in the results section.

3) I request the authors to provide details of all the primers used in this study.

Response: A table- “Table-1 entitled Details of Primers used in study” has been added at the end of the manuscript and the sentence “Details of the primer sequence are given in Table 1.” Has the mention of the same in the main text in the subheading RNA extraction and quantitative reverse transcriptase PCR in the Method section on page 9

4) Can the authors comment on the findings of figure 2D where they found that the amount of IFN is lower than the control in the Doxy IV case? What does this signify?

Response: On Page 13 we have included a justification regarding IFNγ levels in the original manuscript. In the revised manuscript we have included recent Ref. to strengthen our discussion regarding the finding. It reads as-

“IFN-γ is responsible for macrophage activation in alveolar tissue and inducing autophagy. However, increased levels of IFN-γ are considered beneficial when it comes to countering the viral attack, and specifically counter acts LPS induced immune activation by suppressing TLR4-mediated gene activation (Sun et al 2023, Kang et al 2018).”

5) In fig. 3C the authors say fold change on the y-axis, but it needs to be clarified it is with respect to which parameter. I request the authors to clarify it.

Response: The fold change is with respect to the housekeeping gene; HPRT. The figure legend has been appropriately modified to reflect the same (Please see the highlighted part below)

 A) In-Gel Zymography was performed to detect MMP activity in the BALF samples. (B) Quantification of the enzymatic activity was done using Image J software, with results represented as Mean± SEM; n= 5 mice per group; *p < 0.05, as analyzed by one-way ANOVA. (C)-D) qPCR-based relative quantification of MMPs expression in the lungs harvested at day 12 from each treatment group; HPRT was used as the housekeeping gene; results showing fold change with respect to HPRT (the housekeeping gene); are represented as Mean± SEM of n = 5 mice per group, *p < 0.05, **p < 0.01, as analyzed by one-way ANOVA.

6) I request the authors to rearrange the paragraph titled “Doxy NPs decrease immune cell infiltration in BALF”. The current paragraph is not very clear.

Response: As per the reviewer’s suggestion we have rearranged the paragraph by adding paragraph breaks and organizing it as per cell population. We anticipate that it is more in flow and easy to understand.

7) For the benefit of the readers, I request the authors to mention what SiglicFHi stands for and maybe cite relevant work.

Response: Per Reviewer’s suggestion we have defined the Siglicf Hi and cited relevant references in the text. Please see the highlighted version below.

Of particular interest here is the macrophage population defined as SiglicFHi (macrophages that have high Siglic F receptor density) that was particularly affected by doxycycline treatment (Robida et al 2022).

8) In figure 4, the authors have done statistical tests for different groups in different panels. I request the authors keep everything consistent.

Response: Statistical analysis performed for figure 4 is consistent with others. The graphs have been labeled alphabetically and represent different cell populations and not treatment groups. The statistical analysis has been performed amongst different treatment groups using One way ANOVA (GraphPad Prism 9 software) and is represented as Mean ± SEM at p≤0.05

Figure 4A shows the gating strategy. 

Figure 4B-4I – Statistical analysis of Immune cell population in bronchoalveolar lavage fluid amongst different treatment groups.

9) In supplementary fig. S1, I request the authors to add the number of sample information. Also, what does the error bar represents?

Response: Appropriate details have been added, and legends now read as; Fig. S1: Characterization of Doxy Nps

A) Particle size analysis from 3 different batches of Doxy Nps; n=3, results represented as Mean±SD

B) Quantification of Doxycycline released from the Doxy Nps in 24 hrs, n=3, results represented as Mean±SD

C) Average amount of Doxycycline released from 3 different batches of Doxy Nps, n=3, results represented as Mean±SEM

Possible typos:

1) In page 6, I request the authors to use a standard curve instead of std which can be confused with standard deviation.

Response: We have made the appropriate change as per the suggestion.

2) In page 7, the authors mention fig. 3A, but I think it is a typo.

Response: On Page 7 the Figure number has been changed to 4A instead of 3A. Thank you for this observation. 

3) In page 11, I think the authors meant Supp Fig. “S1”.

Response: We have made appropriate changes on page 11 to mention Fig. S1. 

4) The captions in figure 2 should be corrected. I think it should be A to D instead of B to D.

Response: The captions have been corrected.

---

## [Decision Letter · Decision Letter 1]

2 Mar 2023

PONE-D-22-32324R1Elastin targeted nanoparticles delivering Doxycycline mitigate cytokine storm and reduce immune cell infiltration in LPS mediated lung inflammation.PLOS ONE

Dear Dr. Vyavahare,

Thank you for submitting your manuscript to PLOS ONE. After careful consideration, we feel that it has merit but does not fully meet PLOS ONE’s publication criteria as it currently stands. Therefore, we invite you to submit a revised version of the manuscript that addresses the points raised during the review process.

Your manuscript was reviewed by same reviewers and one of the reviewers suggested some minor changes regarding figure. Please address those comments in the revised version. Please note that a quick editorial decision will be taken in the next round without sending to reviewers.

We look forward to receiving your revised manuscript.

Kind regards,

Partha Mukhopadhyay, Ph.D.

Section Editor

PLOS ONE

Journal Requirements:

Note: HTML markup is below. Please do not edit.]

Reviewers' comments:

Reviewer's Responses to Questions

**Comments to the Author**

1. If the authors have adequately addressed your comments raised in a previous round of review and you feel that this manuscript is now acceptable for publication, you may indicate that here to bypass the “Comments to the Author” section, enter your conflict of interest statement in the “Confidential to Editor” section, and submit your "Accept" recommendation.

Reviewer #1: (No Response)

Reviewer #2: All comments have been addressed

2. Is the manuscript technically sound, and do the data support the conclusions?

Reviewer #1: Yes

Reviewer #2: (No Response)

3. Has the statistical analysis been performed appropriately and rigorously? 

Reviewer #1: Yes

Reviewer #2: (No Response)

4. Have the authors made all data underlying the findings in their manuscript fully available?

Reviewer #1: Yes

Reviewer #2: (No Response)

5. Is the manuscript presented in an intelligible fashion and written in standard English?

Reviewer #1: Yes

Reviewer #2: (No Response)

6. Review Comments to the Author

Reviewer #1: The authors have addressed most of the points. I have two more minor comments -

1. Kindly include the justification to comment number 1 (Reviewer 1) in the introduction section to highlight more on the scope of the current study.

2. Kindly rearrange the x- and y- axis of the Figure 4Aiii and 4Aiv. Whatever, justification the authors given to this point, they did not follow this on Figure Av, Avi and Avii. The entire population acquired should be visible. There is no question about the analysis. But, the representation is not adequate. The authors can easily fix this with the help of T button in the FlowJo.

Reviewer #2: (No Response)

7. PLOS authors have the option to publish the peer review history of their article (what does this mean?). If published, this will include your full peer review and any attached files.

Reviewer #1: **Yes: **Abhishek Basu

Reviewer #2: No

---

## [Author Response · Author response to Decision Letter 1]

9 Mar 2023

We again thank the reviewers for their comments and suggestions. 

Reviewer #1: The authors have addressed most of the points. I have two more minor comments -

1. Kindly include the justification to comment number 1 (Reviewer 1) in the introduction section to highlight more on the scope of the current study.

Response: As per Reviewer’ suggestion we have included the justification to comment number 1 (Reviewer 1) in the introduction section to highlight more on the scope of the current study on Page Number 4, in paragraphs 1 and 2.

2. Kindly rearrange the x- and y- axis of the Figure 4Aiii and 4Aiv. Whatever, justification the authors given to this point, they did not follow this on Figure Av, Avi and Avii. The entire population acquired should be visible. There is no question about the analysis. But the representation is not adequate. The authors can easily fix this with the help of T button in the FlowJo.

Response: As per reviewer’ suggestion we have made changes in the Figure4Aiii and 4Aiv and have rearranged the x- and y- axis of the Figure 4Aiii and 4Aiv to include the entire population.

---

## [Decision Letter · Decision Letter 2]

11 May 2023

Elastin targeted nanoparticles delivering Doxycycline mitigate cytokine storm and reduce immune cell infiltration in LPS mediated lung inflammation.

PONE-D-22-32324R2

Dear Dr. Vyavahare,

We’re pleased to inform you that your manuscript has been judged scientifically suitable for publication and will be formally accepted for publication once it meets all outstanding technical requirements.

Kind regards,

Partha Mukhopadhyay, Ph.D.

Section Editor

PLOS ONE

Additional Editor Comments (optional):

Reviewers' comments:

Reviewer's Responses to Questions

**Comments to the Author**

1. If the authors have adequately addressed your comments raised in a previous round of review and you feel that this manuscript is now acceptable for publication, you may indicate that here to bypass the “Comments to the Author” section, enter your conflict of interest statement in the “Confidential to Editor” section, and submit your "Accept" recommendation.

Reviewer #1: All comments have been addressed

Reviewer #2: All comments have been addressed

2. Is the manuscript technically sound, and do the data support the conclusions?

Reviewer #1: Yes

Reviewer #2: (No Response)

3. Has the statistical analysis been performed appropriately and rigorously? 

Reviewer #1: Yes

Reviewer #2: (No Response)

4. Have the authors made all data underlying the findings in their manuscript fully available?

Reviewer #1: Yes

Reviewer #2: (No Response)

5. Is the manuscript presented in an intelligible fashion and written in standard English?

Reviewer #1: Yes

Reviewer #2: (No Response)

6. Review Comments to the Author

Reviewer #1: The authors have addressed all the comments. I have no further comments or advice.

Reviewer #2: (No Response)

7. PLOS authors have the option to publish the peer review history of their article (what does this mean?). If published, this will include your full peer review and any attached files.

Reviewer #1: **Yes: **Abhishek Basu

Reviewer #2: No

---

## [Editor Report · Acceptance letter]

16 May 2023

PONE-D-22-32324R2 

Elastin-targeted nanoparticles delivering doxycycline mitigate cytokine storm and reduce immune cell infiltration in LPS-mediated lung inflammation. 

Dear Dr. Vyavahare:

I'm pleased to inform you that your manuscript has been deemed suitable for publication in PLOS ONE. Congratulations! Your manuscript is now with our production department. 

Kind regards, 

on behalf of

Dr. Partha Mukhopadhyay 

Section Editor

PLOS ONE